# Impact of COVID-19 Confinement on Quality of Life of Patients with Age-Related Macular Degeneration: A Two-Wave Panel Study

**DOI:** 10.3390/jcm12062394

**Published:** 2023-03-20

**Authors:** María R. Sanabria, Paola S. Calles-Monar, Ana M. Alonso-Tarancón, Rosa M. Coco-Martín, Agustín Mayo-Iscar

**Affiliations:** 1Palencia University Hospital Complex, Sanidad Castilla y Leon (SACYL), 34004 Palencia, Spain; 2Instituto de Oftalmobiología Aplicada (IOBA), Facultad de Medicina, Universidad de Valladolid, 47011 Valladolid, Spain; 3RICORS de Enfermedades Inflamatorias, Carlos III Health Institute, 28220 Madrid, Spain; 4Department of Statistics and Operations Research, Instituto de Investigación en Matemáticas (IMUVa), Universidad de Valladolid, 47011 Valladolid, Spain

**Keywords:** health-related quality of life, social support, surveys and questionnaires, age-related macular degeneration, neovascular, COVID-19

## Abstract

Age-related macular degeneration (AMD) is the leading cause of blindness in developed countries. Intravitreal injections of antiangiogenic agents (anti-VEGF) can stop vision loss in the neovascular form of the disease (nAMD). The aim of this study was to assess the general health-related quality of life (HRQoL) in a cohort of patients with nAMD treated with intravitreal anti-VEGF injections and to detesrmine to what extent their HRQoL was affected by COVID-19. This was an observational, analytical, and longitudinal study performed with a two-wave panel survey. Clinical outcomes, HRQoL, and tangible support were evaluated. In the final survey, changes in living conditions and medical visits due to the COVID-19 pandemic were also examined. Of the 102 patients initially interviewed in the before-COVID survey, 24 were lost after 30 months of follow-up. In the initial assessment, the mean health index was 0.73 ± 0.2. The EQ VAS score worsened at the final survey (*p* = 0.048). Patients needing treatment in both eyes (*p* = 0.007) and with lower levels of bilateral visual acuity (*p* = 0.018) reported an increase in social support at the final survey. In conclusion, patients perceived a worsening in HRQoL after confinement. However, patients enjoyed good social support that improved in the after-COVID survey.

## 1. Introduction

Age-related macular degeneration (AMD) is the leading cause of blindness in developed countries, and its prevalence increases with age, reaching 7.1% of people older than 75 years [1]. The neovascular form of the disease progresses quickly, but treatment with intravitreal injections of antiangiogenic agents (anti-VEGF) can stop vision loss [2]. The introduction of anti-VEGF treatment since 2004 has led to a decrease in visual impairment due to neovascular AMD (nAMD), also preventing the deterioration of health-related quality of life (HRQoL) to which patients with nAMD were doomed before this treatment [3]. The prevalence of blindness due to AMD has declined by almost 30% in recent decades [4]. Current intravitreal anti-VEGF therapies require repeated administrations, and optimal results are obtained with monthly or bimonthly injections that must be applied in a theater or in a clean room in the outpatient setting, which involves frequent trips to the clinic [5]. In this sense, the treatment poses a marked inconvenience for the patient and their caregivers, which can in certain ways affect the patient’s HRQoL.

Quality of life (QoL) is a subjective measure that is influenced by various factors, such as expectations, relationships, routines, health, and disability [6]. Self-reported HRQoL assessments provide a comprehensive reflection of the disease impact and guide physicians in the proper care of patients. Since the advent of anti-VEGF treatments, several studies have addressed the relationship between vision-related QoL in nAMD patients treated with anti-VEGF [3,5]. Nevertheless, the measure of vision-related QoL might not reflect the global general health range as nonspecific HRQoL instruments do [3].

Severe acute respiratory syndrome coronavirus 2 (SARS-CoV-2) has spread around the world since December 2019, and the World Health Organization (WHO) declared the COVID-19 pandemic on 11 March 2020 [7]. The government of Spain declared a state of alarm with a nationwide lockdown on March 14, and strict confinement measures lasted until June 2020 [8,9]. From this date on, progressive and slow relaxation of confinement measures was established. Although the COVID-19 pandemic affects the whole population globally, elderly patients, such as those affected by AMD, suffer worse outcomes with SARS-CoV-2 infection [10]. Confinement increased psychological discomfort and feelings of uncertainty and concern about suffering or contracting a serious illness, especially in older people [11]. Although most scientific ophthalmological societies recommended maintaining the administration of intravitreal injections in patients with nAMD during confinement [12,13], some patients interrupted the therapy by choice out of fear of SARS-CoV-2 or because it was impossible to reach the hospital [14]. In these ways, the pandemic has added concern to previously distressed patients [10].

Therefore, the aim of this study was to evaluate the general HRQoL in a cohort of nAMD patients treated with intravitreal anti-VEGF injections and to determine to what extent their QoL has been affected by the COVID-19 pandemic.

## 2. Materials and Methods

This was an observational, analytical, and longitudinal panel study. The Medical Ethical Committees of Complejo Asistencial Universitario de Palencia (CAUPA) approved the design of the study according to Spanish laws and the Declaration of Helsinki including its subsequent amendments.

### 2.1. Setting

This study was carried out at CAUPA. This secondary-care hospital has certain tertiary care services that belong to the universal national health system and provides free healthcare to residents in Palencia Province (161.321 inhabitants) with a wide geographical dispersion [15]. The percentage of inhabitants aged 65 and over in Palencia represents a quarter of the population of the province (25.99%) [16]. In 2019, more than 3500 intravitreal injections were administered in the ophthalmology service of CAUPA.

The present study consisted of a two-wave panel survey. For the initial survey, consecutive sampling was carried out from 1 November 2018 to 30 November 2018. The interview was personal, and the patients were informed about the purpose of the study and signed an informed consent form. Thirty months after the initial survey, in March 2021, the previously recruited patients in 2018 were contacted by phone, and the questionnaires were repeated, adding some questions related to changes in intravitreal treatment during the home confinement and postconfinement periods. Two weeks before the telephone interview, a letter was sent to the patients announcing the upcoming call and reminding them of their voluntary participation in the survey. Clinical data related to this second survey were retrieved from the clinical files of each patient corresponding to the last visit closest to the data of the phone interview. In February 2021, the third wave of COVID-19 was especially aggressive in Palencia Province, shooting up to a cumulative incidence of more than 1000 positive cases per 100,000 inhabitants in one week [17].

### 2.2. Inclusion and Exclusion Criteria

Patients who had been diagnosed with nAMD under anti-VEGF treatment for at least 1 year at the first interview and attended the Ophthalmology Service of the Complejo Asistencial Universitario de Palencia (CAUPA) were included in the study. Patients with significant lens opacities, glaucoma, diabetic retinopathy, and other comorbidities were excluded in the initial survey. Patients with cognitive impairment or not wishing to participate were also excluded.

Patients included in the first personal query who could not be entered into the second telephone query were asked for their reason for not responding.

### 2.3. Study Outcomes

We used the following study outcomes grouped into four domains of interest:Clinical outcomes: visual acuity (VA), visual impairment (WHO definitions) [18], number of intravitreal injections and unilateral or bilateral disease, and cataract surgery between the two surveys.HRQoL: Patients were asked to complete the EQ-5D-3L questionnaire [19]. The validated Spanish version of the EQ-5D-3L questionnaire, including the EQ Visual Analogue Scale (EQ VAS), was used [19]. The EQ-5D instrument is a generic instrument that provides a summary of HRQoL. The questionnaire was developed by the EuroQol Group for measuring HRQoL and consists of the EQ-5D descriptive system and the EQ VAS. The EQ descriptive system explores five health dimensions (mobility, self-care, activity, pain/discomfort, and anxiety/depression) that produce a five-digit health state profile for each patient. EQ-5D health states may be converted afterwards into a definite summary number: an index value also known as the health index. The health index reflects how good or bad a health state is, according to the preferences of the general population of a precise country/region, ensuring that the index values represent the societal perspective. The index value has a maximum value of 1 for perfect health, 0 represents death, and negative values reflect states worse than death [19]. The EQ VAS is a visual scale from 0 to 100, where 0 is the worst imagined state of health and 100 means the best [19].Social support: To assess the caregiver support and social network that the patient can have if needed, four questions concerning tangible support from the Medical Outcomes Study (MOS) Social Support questionnaire were used [20]. The MOS Social Support Survey (MOS SSS) Instrument consists of four separate social support subscales or dimensions (emotional/informational support, tangible support, affectionate support, and positive social interaction). A higher score on the scale or for overall support indicates more support. The instrumental or tangible support subscale qualifies the type of material or assistance aid, which can be measured in some way. This includes economic or financial support, material help in obtaining goods or services, collaboration with housework, and caring for the patient [20]. Responses range from 1 (none of the time) to 5 (all of the time). Higher scores indicate a higher level of social support. The maximum possible tangible scale score is 20. Scale scores were transformed to a 0–100-scale tangible support index for better comparison [21].Changes in living conditions and medical visits due to the COVID-19 pandemic. A specific questionnaire was designed and offered to the patients to determine pandemic-related questions.

### 2.4. Variables

Demographic and medical variables were retrieved from the clinical files of patients. The data included information about the patients’ sex, age, treated eye/s, unilateral and bilateral distance best-corrected VA (BCVA), degree of visual impairment, unilateral or bilateral need for treatment, time from the beginning of treatment, number of intravitreal injections, change in VA after starting treatment, and change in VA between surveys. All-distance best-corrected VAs (BCVAs) were recorded using a Snellen chart and converted to the logarithm of the minimum angle of resolution (LogMAR) using a validated procedure [22]. BCVA and visual impairment grade were considered at both the initial (pre-COVID) and final query (after COVID). The VA change per eye was considered an improvement or worsening if there was an increase or decrease in one step on the LogMAR scale, respectively. Regarding the VA change per patient, improvement was considered to occur when there was improvement in both eyes or when the combined change in VA in both eyes was an improvement. Worsening was considered when there was worsening in both eyes or when the combined change in VA in both eyes worsened.

Information about the distance from the residence to the ophthalmology service was also recorded.

### 2.5. Statistical Analysis

Numerical variables were summarized as means and standard deviations, and categorical variables were summarized as frequencies and percentages. The 95% confidence interval (95% CI) was calculated for the corresponding parameters. Numerical variables such as age, treatment duration, number of injections, and VA were set in quartiles. *t*-tests, one-way analysis of variance, or correlation coefficients were used to relate QoL scores and social support scores to sociodemographic and clinical variables.

EQ-5D results were compared with similar data from an age-matched population based on the Spanish 2012 National Health Survey. EQ-5D results were also compared with those from an age-matched cohort divided by sex based on the same survey. The Spanish 2012 National Health Survey was the last National Health Survey that used the EQ-5D-3L questionnaire.

*p* values lower than 0.05 were considered statistically significant. The statistical analysis was performed using the R-4.1.0 package (R Foundation for Statistical Computing, Vienna, Austria).

## 3. Results

### 3.1. Study Population

Of the 102 patients initially surveyed, 24 did not answer the final survey, including 4 patients who decided to abandon the treatment and ophthalmic periodic visits, 6 patients who were not located, 4 patients who refused to participate, and 10 patients who died (2 of them died due to COVID-19-related complications).

Seventy-eight patients were interviewed in the two waves of the study. The mean age of the patients at the initial survey was 81.2 ± 7.1 years, and 45 were female (57.7%). The demographic data of the patients are detailed in Table 1.

Patients had to travel a mean of 37.2 ± 48 [95% CI: 26.2–48] km at the initial survey and 45.3 ± 94.3 [95% CI 24.1–66.6] km at the final survey. No patient participating in the study resided in a nursing home at the initial survey. Three patients who lived alone at the initial survey changed their domicile: two patients went to live in a nursing home, and one went to live with family.

In the lapse of time between both surveys, nine eyes of nine patients developed significant lens opacities and were operated on. No patient developed uncontrolled ocular hypertension or any other retinal complication.

### 3.2. Study Outcomes

Table 1 shows the clinical outcomes of the study. The patients were under intravitreal treatment for a mean of 44 months at the time of the initial survey and 74 months at the final survey. At the final survey, 66 patients (84.6%) had continued treatment, 12 had suspended treatment, 8 (10.3%) had inactive disease for more than 12 months, and 4 (5.1%) had a nonresponding and/or poor prognosis.

Before starting antiangiogenic treatment, 50 (64.1%) of our patients had no visual impairment, 12 (15.5%) had mild impairment, 11 (14.1%) had moderate impairment, 3 (3.8%) had severe impairment, and 2 (2.56%) were blind.

After a mean follow-up time of 74.6 months, the mean BCVA of treated eyes declined during the study from the pretreatment LogMAR BCVA of 0.58 ± 0.42 to 0.77 ± 0.83 at the final survey. Although at the initial survey 39 patients (50%) were receiving treatment in both eyes, by the time of the final survey, 53 patients (67.9%) required bilateral treatment.

In the initial assessment of HRQoL with the EQ-5D-3L questionnaire, the mean health index was 0.73 ± 0.2 and the EQ VAS was 74.5 ± 19.0. The final survey results of EQ-5D-3L were 0.70 ± 0.2 in the health index and 70.1 ± 16.5 in the EQ VAS. This last result was worse than that obtained in the first query (*p* = 0.048). In the initial survey, the two most frequent health problems were “pain/discomfort” (in 69.23% of patients) and “mobility problems” (38.46%). In the final survey, the two most frequent health problems were “pain/discomfort” and “problems with usual activities” in 66.67% and 48.87% of patients, respectively.

Detailed results of both the initial and final assessment of QoL (EQ-5D-3L) in the five-dimension descriptive system are shown in Table 2. The descriptive system related to “self-care” and “usual activities” worsened in the final after-COVID interview (*p* = 0.005 and 0.01, respectively).

Comparing the results for HRQoL of the present sample with an age-matched sample based on the Spanish 2012 National Health Survey, we found significant differences in the pre-COVID assessment of “self-care”, “usual activities”, and “pain discomfort”, which were better in our sample. Nevertheless, there were no differences in mean scores in any descriptive system between the Spanish 2012 National Health Survey and the final survey except for “pain and discomfort”, which was worse in our sample (Table 2). Figure 1 shows a descriptive graphic comparing the present sample with an age-matched sample separated by sex, based on the Spanish 2012 National Health Survey.

We found evidence for an interaction effect between the EQ-5D and VAS scores and the following variables: sex, age, receipt of treatment in both eyes, visual impairment, and bilateral visual acuity in both the initial and final surveys (Table 3).

The mean score of the MOSS SSS tangible social support subscale was 17.9 ± 3.5 in the initial survey and 18.6 ± 3.3 in the final survey (*p* = 0.062). An improvement was found in the questions corresponding to social tangible support in the cases of “needing help if confined to bed” or “preparing meals” when comparing the initial and final surveys. Detailed results of the tangible support subscale in the two waves of the study are shown in Table 4. Significant differences were observed between the tangible support score in the initial survey and the following variables: sex, living conditions, situation in relation to treatment, and change in bilateral BCVA from the beginning of the treatment to the initial survey (Table 5). In the final survey, those differences did not reach statistical significance. The score change for tangible support between the initial and final surveys was not associated with any variable except in the case of patients needing treatment in both eyes (*p* = 0.007) and patients with lower levels of bilateral BCVA (*p* = 0.018), which increased the support at the final survey.

Regarding the pandemic-related questions, 6 out of the 78 patients had experienced and recovered from coronavirus infection. Only 20 (25.6%) patients reported having delayed one of their appointments, and only 6 of them had delayed their appointments by more than 1 month (7.7%). In five cases, delayed appointments were requested by the patient: one for being in quarantine and in four cases because of problems getting to the hospital. Subjectively, most of the patients (85.9%) did not feel that their maculopathy had worsened from the beginning of the pandemic.

## 4. Discussion

Here, we present the results of a long-term, real-world study that shows the effects of intravitreal treatment in a sample of nAMD patients by assessing a broad set of endpoints: long-term clinical outcomes, QoL, and social tangible support. The study reveals that patients treated with AMD experience a decrease in quality of life over time. However, there is an important peculiarity which is that during the period that passed between the first and the second interview they suffered the consequences of a pandemic that dramatically changed their lives. All our patients were affected by the pandemic in many ways because there could have been delayed appointments or they may have had difficulties reaching the hospital not only for their eye problem but also other medical problems. Moreover, many of them loss contact with family, changed their routines, were threatened or affected by the SARS-Cov-2 infection, etc. In the present study, we focused our attention on QoL and tangible support with validated methods of measurement under the described circumstances.

The clinical outcomes of our sample reflect what has been described in other real-world, long-term studies on the subject: there is a gradual deterioration of visual function over time in nAMD patients, despite the high mean number of intravitreal injections [23,24]. There are several possible reasons for this in the present study. The patients of our sample were treated for nearly 7 years at the final survey, and it has been reported in several real-world observational studies that patients followed for more than 7 years have a vision decrease that can reach a loss of 4.3 letters per year [25]. Furthermore, visual loss in the long term has been related to progression to atrophy or suboptimal therapy, as was precisely the case in many patients that delayed appointments and injections in our sample [25].

Our sample also showed a progressive increase in bilateral treatments due to sequential involvement of the fellow eye, which was expected as this is a bilateral condition [26].

Although the influence of anti-VEGF treatment on vision-related QoL has been assessed in several studies, there was a shortage of information about how repeated hospital visits and frequent anti-VEGF injections influence other vital facets of patients’ functionality that can be measured through generic instruments such as EQ-5D. Recently, Finger et al. [3] published a systematic review on the impact of anti-VEGF in nAMD on patient outcomes and highlighted this fact. Good HRQoL is not synonymous with good QoL. Generic measures can underestimate some of the health effects of certain conditions such as visual problems [26]. Most of the domains explored with generic instruments are not directly modified by visual impairment (e.g., pain, energy), although they can be affected indirectly (e.g., motility, depression). Obviously, the high frequency of problems related to pain/discomfort found in our sample is not related to AMD but the questionnaire reflects other health problems of our AMD patients. Pain/discomfort is a frequent complaint in the general population, especially in women (Figure 1). Nevertheless, the generic measures of HRQoL such as EQ-5D have some advantages, such as measuring the influence of conditions or their treatments on global HRQoL and offering the possibility to compare this impact within different disorders. To improve the sensitivity of generic instruments in relation to visual problems, some authors advocate adding a bolt-on vision dimension to the EQ-5D. This bolt-on dimension consists of adding to the five standard dimensions one more dimension in which the patient evaluates his or her visual difficulties by choosing between these response options: “no problems”, “some problems”, or “extreme problems” in seeing [27]. Although vision is underrepresented in EQ-5D, our study also collected data regarding visual impairment and VA that are directly correlated with vision-related QoL and could complete this gap [28].

Concerning the questionnaire used to explore social support, we did not use the whole MOS social support survey (19 questions) in order to minimize the potential for survey fatigue. Because of the possibility of intervention, we were interested in having information about the provision of material aid or behavioral assistance from others that the patient could need because of his or her ocular disease.

The study design—panel study—allows tracking the changes that have taken place over time and, in this case, with the special impact of the pandemic. Nevertheless, there are two main drawbacks in this kind of study: loss of participants and reactivity. Loss to follow-up (also called attrition) and exclusion criteria (such as cognitive impairment) could have biased the results. Selective attrition—loss to follow-up, especially in older and sicker patients—is one of the main problems in long-term studies with very old populations [29,30]. In the present study, attrition accounted for less than 25% of the initial sample, which is much less than that reported in other studies of similar follow-up, which can reach up to 50% in studies of more than 6 years [22,27,28]. Reactivity accounts for the bias in opinion that occurs when the same questions are posed to individuals several times. However, in the present study, the interval between interviews was 30 months, which is expected to be enough time to minimize reactivity.

The present sample was compared to the population of the age-matched Spanish 2012 National Health Survey. Perceived health status has improved since 2012 [31,32]. This could explain, at least in part, the fact that our sample showed higher mean HRQoL scores [33]. Other studies published by Péntek et al. [34] also found no differences compared to the age-matched general population. In contrast, there are many publications underlining the negative effects of AMD on QoL [5]. However, most published studies in this field lack discrimination between neovascular and non-neovascular AMD or were performed before the anti-VEGF era [5], and it has already been mentioned that anti-VEGF treatment yields an important improvement in the QoL of nAMD patients [3].

In our sample, females and older patients showed lower QoL scores in the initial and final surveys, which is a common finding in QoL studies [35,36]. Receiving treatment in both eyes was correlated with worse scores on the EQ-5D, but the differences did not reach statistical significance in the final survey. Lower visual impairment and, especially, higher bilateral VA were related to higher scores on QoL questionnaires, which was expected and in accordance with Péntek et al.’s [34] findings.

The mean EQ-5D score did not change substantially after confinement, but a detailed reading of the data shows that scores on self-care problems and usual activities worsened, as shown in the 2012 figures. Most patients who reported worsening in these questions referred to having problems washing and dressing themselves (self-care) or having some problems in their usual activities, as can be seen in Table 2. Physical activity decreased significantly during confinement and sedentary time increased. This could have accelerated the usual decline in muscle mass, strength, and function in the older adults, especially affecting activities such as “self-care” and “usual activities” [37]. Moreover, in 17.9% of our patients, the second eye was affected in the final survey, worsening the results of this item. Although HRQoL has improved substantially in the last decade in some sense, it has suffered a setback of 10 years due to the COVID-19 pandemic [33].

In general, our patients enjoyed strong tangible or material support. Nevertheless, patients living alone and female patients had lower scores in support and should be surveyed by social workers. The social isolation recommended for COVID-19 could have had harmful consequences for patients having difficulties attending medical appointments due to the cancellation of public transport, the impossibility of being accompanied to the hospital, or concerns about COVID-19. Delayed medical care for chronic medical conditions has been widely reported during the COVID-19 pandemic, not only during the confinement period but also persisting until 2021 [38,39]. Patients with disabilities or multiple medical conditions most commonly reported a delay in or the avoidance of any medical care [34]. In our study, patients who especially experienced more support were those who maintained treatment or had lower bilateral VA, and they were more aware of this support in the final survey. In the same way, visual impairment made our patients especially vulnerable by reducing their independence required for attending their visits.

Treatment delay due to the COVID-19 pandemic has been limited in our department in comparison with other settings, as anti-VEGF treatment was guaranteed for nAMD patients and appointments were infrequently deferred [40]. However, it is concerning that certain patients who lived far away from the hospital could not attend their appointments due to the cancellation of public transportation services or not having anyone who could bring them to the hospital.

The present study has some limitations. It involved a sample of nAMD patients from a unique hospital with geographical and socioeconomic peculiarities that could make generalization of the conclusions difficult. Nevertheless, this could be comparable to the difficulties encountered in similar regions with other very old or sparse populations, the number of which are increasing, particularly in Spain and generally in Europe. The bias implicit in the panel study design has already been mentioned. Despite these drawbacks, this study shed light on many aspects of the impact of anti-VEGF on patients’ lives.

The present study highlights nAMD patients’ needs, considering the worsening HRQoL, especially in self-care and motility problems that should be addressed by health authorities, particularly for those patients we have identified to have less support. Finally, Spanish and European health stakeholders are challenged to meet the specific needs of regions with an aging and dispersed population, such as ours.

## 5. Conclusions

Patients under intravitreal treatment in our study usually enjoyed good social support. In general, HRQoL measured with generic instruments was not worse in patients with nAMD than in the general population, although females, older patients, and patients with bilateral disease leading to poor bilateral VA and therefore greater visual impairment had worse markers. Patients perceived a worsening in QoL after confinement.

## Figures and Tables

**Figure 1 jcm-12-02394-f001:**
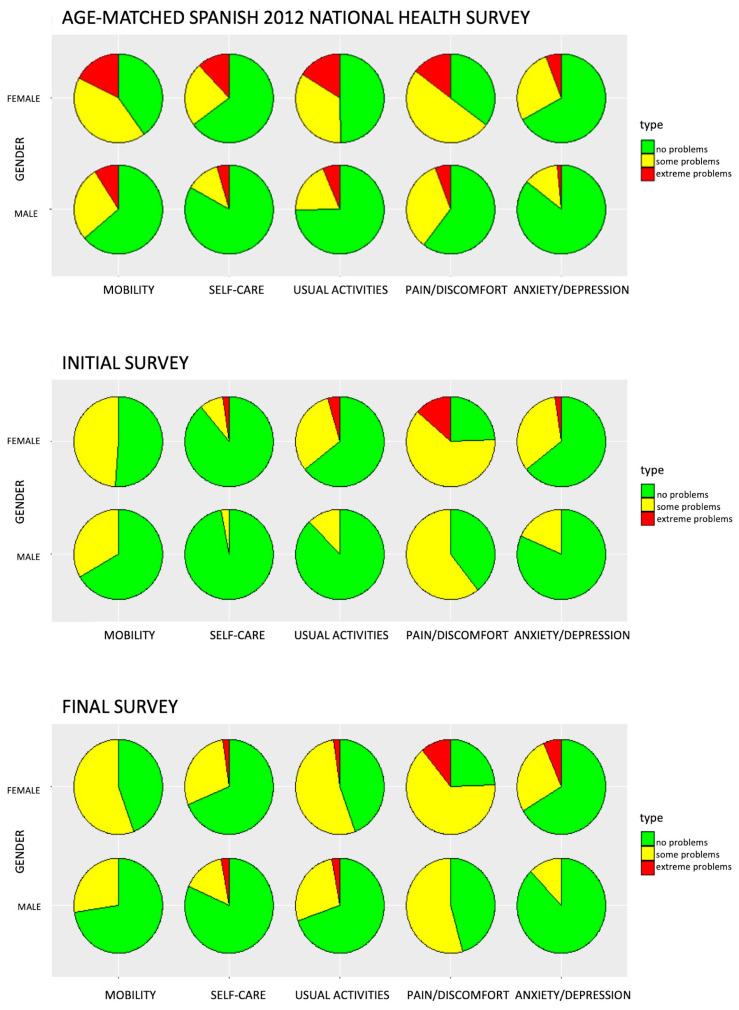
EQ-5D-3L results compared with age-matched results divided by sex based on the Spanish 2012 National Health Survey. Results are shown by dimensions. Level 1 in light gray = no problems. Level 2 in white = some problems. Level 3 in dark gray = severe problems.

**Table 1 jcm-12-02394-t001:** Demographic characteristics of the sample at the initial and final surveys.

	Mean ± SD [%] (95% CI)	
	Initial Survey	Final Survey
Age (years)	81.2 ± 7.1 (79.6–82.8)	83.2 ± 7.1 (81.6–84.8)
Sex		
Male	33 [42.3%] (31.2–54.0)	
Female	45 [57.7%] (46–68.8)	
Living conditions:		
The patient lives alone	23 [29.5%] (19.7–40.9)	20 [25.6%] (16.4–36.8)
The patient lives with his or her family	54 [69.2%] (57.8–79.2)	55 [70.5%] (59.1–80.3)
The patient lives in a nursing home	0 [0%] (0–4.6)	2 [2.6%] (0.3–9.0)
Other (the patient lives with a friend)	1 [1.3%] (0–6.9)	1 [1.3%] (0.0–6.9)
Time from the beginning of the treatment (months)	44.6 ± 26.6 [38.7–50.6]	74.6 ± 27.1 [67.9–81.3]
Number of intravitreal injections applied per patient	23.4 ± 13.3 [20.4–26.4]	38.5 ± 17.3 [34.6–42.4]
BCVA of treated eyes—LogMAR	0.66 ± 0.61 [0.55–0.77]	0.77 ± 0.83 [0.63–0.92]
Visual impairment *		
No impairment (≤0.3)	52 [66.7%] (55.1–76.9)	52 [66.7%] (55.1–76.9)
Mild impairment (>0.3–≤0.5)	18 [23.1%] (14.3–34)	13 [16.7%] (9.2–26.8)
Moderate impairment (>0.5–≤1)	6 [7.7%] (2.9–16)	9 [11.5%] (5.4–20.8)
Severe impairment (1–1.3)	2 [2.6%] (0.3–9)	2 [2.6%] (0.3–9)
Blindness (>1.3)	0 [0%] (0–4.6)	2 [2.6%] (0.3–9)
BCVA changes after treatment (eyes):		
Improvement	45 [38.5%] 29.6–47.9	57 [43.5%] 34.9–52.4
No change	26 [22.2%] 15.1–30.8	16 [12.2%] 7.1–19.1
Worsening	46 [39.3%] 30.4–48.8	58 [44.3%] 35.6–53.2

SD: standard deviation; [%]: percentage; (95% CI): 95% confidence interval. * Visual impairment as defined by the World Health Organization [17]: No impairment = LogMAR visual acuity ≤ 0.3; Mild impairment = LogMAR visual acuity >0.3–≤0.5; Moderate impairment = LogMAR visual acuity >0.5–≤1; Severe impairment = LogMAR visual acuity 1–1.3; or blindness (>1.3). BCVA: best-corrected visual acuity. The change in visual acuity per eye was considered an improvement or worsening if there was an increase or decrease in one step on the LogMAR scale, respectively.

**Table 2 jcm-12-02394-t002:** EQ-5D-3L numbers and frequencies by dimension and level in initial and final surveys.

EQ-5D-3L	Mobility Problems		Self-Care		Problems with Usual Activities		Pain/Discomfort		Anxiety/Depression	
	Initial	Final	Initial	Final	Initial	Final	Initial	Final	Initial	Final
Level 1 (no problems)	48 [61.53%]	44 [56.41%]	72 [92.3%]	58 [74.36%]	58 [74.45%]	43 [55.12%]	24 [30.76%]	26 [33.33%]	56 [71.79%]	59 [75.64%]
Level 2 (some problems)	30 [38.46%]	34 [43.58%]	5 [6.41%]	18 [23.08%]	17 [21.79%]	33 [42.31%]	48 [61.53%]	47 [60.26%]	21 [26.92%]	16 [20.51%]
Level 3 (severe problems)	0 [0%]	0 [0%]	1 [1.28%]	2 [2.56%]	2 [2.56%]	2 [2.56%]	6 [7.69%]	5 [6.41%]	1 [1.28%]	3 [3.85%]
Total	78 [100%]	78 [100%]	78 [100%]	78 [100%]	78 [100%]	78 [100%]	78 [100%]	78 [100%]	78 [100%]	78 [100%]
Any problem	30 [38.46%]	34 [43.58%]	6 [7.69%]	20 [25.64%]	19 [24.36%]	35 [48.87%]	54 [69.23%]	52 [66.67%]	22 [28.20%]	19 [24.36%]
Change in any problem	*p* = 0.63		***p* = 0.005**		***p* = 0.01**		*p* = 0.86		*p* = 0.72	
Any problem: Spanish 2012-NHS	49.27%		27.39%		39.27%		53.85%		24.81%	
* Comparison of our sample with the Spanish 2012-NHS population (*p* value)	0.072	0.374	**<0.001**	0.841	**0.008**	0.369	**0.008**	**0.029**	0.563	1

EQ-5D-3L numbers and frequencies by dimension and level in initial and final surveys. Any problem = levels 2 + 3. Proportions reporting levels within the EQ-5D dimensions of an age-matched population based on the Spanish 2012 National Health Survey (NHS). * Comparison of EQ-5D-3L for the any-problem results of our sample with an age-matched population based on the Spanish 2012 National Health Survey (NHS). Bold denotes significant values.

**Table 3 jcm-12-02394-t003:** Significant results of univariate comparison between demographic and clinical variables with HRQoL quantified by EQ-5D and VAS.

	Initial Survey		Final Survey	
	VAS	Health Index	VAS	Health Index
Sex				
Male	82.090	0.810	73.180	0.790
Female	68.910	0.670	67.780	0.650
*p* value	**0.002**	**0.001**	0.155	**0.009**
Age (years)				
Q1: 64–77	82.380	0.850	78.120	0.870
Q2: 78–82	74.090	0.700	70.430	0.710
Q3: 82–86	70.210	0.780	69.290	0.690
Q4: 87–97	72.200	0.660	65.000	0.610
*p* value	0.114	**0.014**	**0.017**	**0.000**
Receipt of treatment				
One eye	79.670	0.770	72.820	0.730
Both eyes	69.310	0.690	67.310	0.690
*p* value	**0.015**	**0.041**	0.142	0.431
Visual impairment				
No impairment (≤0.3)	75.560	0.770	72.880	0.740
Mild impairment (>0.3–≤0.5)	72.440	0.660	65.380	0.660
Moderate impairment (>0.5–≤1)	67.000	0.600	65.450	0.680
Severe impairment (1–1.3)	87.500	0.720	52.500	0.430
*p* value	0.700	**0.022**	**0.026**	0.077
Bilateral BCVA				
Q1: 0–0.1	87.250	0.880	76.670	0.860
Q2: 0.2	73.000	0.760	76.110	0.760
Q3: 0.3–0.4	73.630	0.750	70.320	0.690
Q4: 0.5–1.3	72.350	0.650	61.250	0.620
*p* value	0.138	**0.002**	**0.003**	**0.004**

HRQoL: health-related quality of life; EQ-5D: EQ-5D-3L questionnaire. Health index: result of EQ-5D-3L. VAS: visual analogue scale. Range of possible scores: 0–100. Visual impairment using World Health Organization definitions [17]. Q1: first quartile; Q2: second quartile; Q3: third quartile; Q4: fourth quartile. BCVA: best-corrected visual acuity. Bold denotes significant values.

**Table 4 jcm-12-02394-t004:** Results of the MOS SSS tangible support subscale: comparison between the initial and final surveys.

	*n* [%] (95% CI)	
	Initial Survey	Final Survey
Availability of someone to help you if you are confined to bed	Mean score *: 4.3 ± 1.3	Mean score *: 4.6 ± 0.9
1 None of the time	5 [6.4%] (2.1–14.3)	1 [1.3%] (0.0–6.9)
2 A little of the time	7 [9.0%] (3.7–17.6)	5 [6.4%] (2.1–14.3)
3 Some of the time	5 [6.4%] (2.1–14.3)	2 [2.6%] (0.3–9.9)
4 Most of the time	6 [7.7%] (2.9–16.0)	7 [9.0%] (3.7–17.6)
5 All the time	55 [70.5%] (59.1–80.3)	63 [80.8%] (70.3–88.8)
*p* value	**0.009**	
Availability of someone to take you to the doctor if you need it	Mean score *: 4.7 ± 0.7	Mean score *: 4.6 ± 0.9
None of the time	0 [0%] (0–4.6)	2 [2.6%] (0.3–9.0)
A little of the time	3 [3.8%] (0.8–10.8)	2 [2.6%] (0.3–9.0)
Some of the time	3 [3.8%] (0.8–10.8)	3 [3.8%] (0.8–10.8)
Most of the time	5 [6.4%] (2.1–14.3)	8 [10.3%] (4.5–19.2)
All the time	67 [85.9%] (76.2–92.7)	63 [80.8%] (70.3–88.8)
*p* value	0.288	
Availability of someone to prepare your meals if you are unable to do it yourself	Mean score *: 4.4 ± 1.1	Mean score *: 4.7 ± 0.9
None of the time	3 [3.8%] (0.8–10.8)	1 [1.3%] (0.0–6.9)
A little of the time	4 [5.1%] (1.4–12.6)	5 [6.4%] (2.1–14.3)
Some of the time	7 [9.0%] (3.7–17.6)	1 [1.3%] (0.0–6.9)
Most of the time	7 [9.0%] (3.7–17.6)	5 [6.4%] (2.1–14.3)
All the time	57 [73.1%] (61.8–82.5)	66 [84.6%] (74.7–91.8)
*p* value	**0.019**	
Availability of someone to help with daily chores if you were sick	Mean score *: 4.5 ± 1.1	Mean score *: 4.7 ± 0.9
None of the time	1 [1.3%] (0.0–6.9)	1 [1.3%] (0.0–6.9)
A little of the time	7 [9.0%] (3.7–17.6)	5 [6.4%] (2.1–14.3)
Some of the time	6 [7.7%] (2.9–16.0)	1 [1.3%] (0.0–6.9)
Most of the time	5 [6.4%] (2.1–14.3)	5 [6.4%] (2.1–14.3)
All the time	59 [75.6%] (64.6–84.7)	66 [84.6%] (74.7–91.8)
*p* value	0.114	

MOS SSS: Medical Outcomes Study Social Support Survey. *n* = number of patients answering each category; [%] = percentage; (95% CI) = 95% confidence interval. * Mean score in each question. The maximum possible score for each question is 5. Bold denotes significant values.

**Table 5 jcm-12-02394-t005:** Univariate comparison between demographic and clinical variables with the MOS SSS tangible support subscale: comparison between the initial and final surveys.

	Initial Survey	Final Survey
	Subscale score (index)	
Sex		
Male	19.09 (94.31)	19.24 (95.25)
Female	17.02 (81.37)	18.11 (88.19)
*p* value	**0.009**	0.136
*p* value *	0.207	
The patient lives		
Alone	16.22 (76.38)	16.35 (77.19)
With his or her family	18.57 (91.06)	19.52 (97)
In a nursing home		20.00 (100)
With friends	20.00 (100)	
*p* value	**0.007**	**0.000**
*p* value *	0.442	
Situation		
Keep on treatment	18.18 (88.63)	18.80 (92.5)
Keep on visiting, with no need of treatment	17.12 (82)	18.50 (90.63)
Keep on visiting, without treatment	14.75 (67.19)	15.25 (70.32)
*p* value	**0.049**	0.067
*p* value *	0.791	
Receive treatment		
One eye	17.95 (87.18)	17.94 (87.12)
Both eyes	17.85 (86.56)	19.12 (94.50)
*p* value	0.898	0.166
*p* value *	**0.007**	
Bilateral BCVA		
Q1 0–0.1	18.00 (87.50)	17.57 (84.82)
Q2 0.2	18.29 (89.31)	17.33 (83.32)
Q3 0.3–0.4	18.33 (89.56)	19.62 (97.63)
Q4 0.5–1.3	17.15 (82.19)	19.46 (97.63)
*p* value	0.337	**0.013**
*p* value *	**0.018**	

The maximum possible tangible subscale score is 20. Scale scores were transformed to a 0–100-scale tangible support index (shown in brackets). * Comparison between the mean scores and tangible support index and the demographic/clinical variables of the initial and final surveys. BCVA: LogMAR best-corrected visual acuity. Bold denotes significant values.

## Data Availability

The data that support the findings of this study are available from the corresponding author upon reasonable request.

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
