# Peer review of "Impact of COVID-19 Confinement on Quality of Life of Patients with Age-Related Macular Degeneration: A Two-Wave Panel Study"

_jcm, 2023, doi:10.3390/jcm12062394_

Round 1

Reviewer 1 Report

The changes of score before and after COVID-19 used by EQ-5D, EQ-VAS, and MOS SSS. This report is very interesting and significant. However, I have a few questions.

If the study of this is discussing the impact of COVID-19, wouldn't it be better to compare those who were affected by the pandemic (e.g., delayed appointments) with those who were not, etc.? Otherwise, it could simply be a decrease in quality of life that the patients treated with AMD feel over time. Or, it would be better to compare this with the previously reported simple change over time.

1, Table 2:  The comparison of Initial study and Final study results for each item is very confusing. Can you devise a way to write the Table?

2, page 6 Line 220:  How do you think are the reasons for the change in "self-care" and "usual activities"? Is the worsening of scores really related to COVID? I think it would be good to add this to the Discussion.

3, Figure 1:  This figure is related to the data in Table 2, but it is disconcerting that gender is suddenly separated here when it was not included in Table 2.

4, page12 Line 363-364: What exactly is "many aspects"?

5, page12 Line 366-367: What exactly do you mean by 'nAMD patients' needs?

Author Response

REVIEWER 1

The changes of score before and after COVID-19 used by EQ-5D, EQ-VAS, and MOS SSS. This report is very interesting and significant. However, I have a few questions.

If the study of this is discussing the impact of COVID-19, wouldn't it be better to compare those who were affected by the pandemic (e.g., delayed appointments) with those who were not, etc.? Otherwise, it could simply be a decrease in quality of life that the patients treated with AMD feel over time decrease in quality of life that the patients treated with AMD feel over time. Or, it would be better to compare this with the previously reported simple change over time.

Answer:            I agree with the reviewer in the appreciation that the work reveal a decrease in quality of life that the patients treated with AMD feel over time, but with an important peculiarity: during the period that has passed between first and second interview there has been a pandemic that has change dramatically the lives of each of us.

This paragraph has been added in the discussion. Page 10 line 280-289

The study reveals a decrease in quality of life that the patients treated with AMD feel over time, but with an important peculiarity: during the period that has passed between first and second interview there has been a pandemic that has change dramatically the lives of each of us. All our patients were affected by the pandemic in many aspects of their lives, not only because there could have delayed appointments or difficulties for reaching the hospital for their eye problem, but also for any other medical problem that they could have. Moreover, many of them lose contact with family, change their routines, were threaten or affected by the infection etc. In present work, we focalize our attention in QoL and tangible support with validated methods of measurement.

1, Table 2:  The comparison of Initial study and Final study results for each item is very confusing. Can you devise a way to write the Table?

Answer:            The tables1 and 2 have been rewritten. We apologize for the inconvenience, but when the manuscript was sent to the web page of the journal and was adapted to the journal template, some of the information of the tables change in line making the information understandable. We have made some changes to avoid this problem. We hope with present reviewed manuscript; the problem had been solved.

2, page 6 Line 220:  How do you think are the reasons for the change in "self-care" and "usual activities"? Is the worsening of scores really related to COVID? I think it would be good to add this to the Discussion.

Answer: The next text has been added in the discussion with the correspondent reference (Page 12 Line 357-364
Most of the patients that denote worsening in these questions refer having problems washing and dressing themselves (self-care) o having some problems in their usual activities, as can be seen in table 2. Physical activity decrease significantly during confinement and sedentary time increase. This could accelerate the usual decline in the muscle mass, strength, and function in the older adults affecting specially to activities as "self-care" and "usual activities". Moreover, in …patients, the second eye was affected in the final survey worsening the results of this items.

Castañeda-Babarro A, Arbillaga-Etxarri A, Gutiérrez-Santamaría B, Coca A. Physical Activity Change during COVID-19 Confinement. Int J Environ Res Public Health. 2020 Sep 21;17(18):6878. doi: 10.3390/ijerph17186878. PMID: 32967091; PMCID: PMC7558959.

3, Figure 1:  This figure is related to the data in Table 2, but it is disconcerting that gender is suddenly separated here when it was not included in Table 2.

Answer: We found an interaction effect between the EQ-5D and VAS scores and sex (page 7 line 238-240). We consider that adding more information in the form of numbers would lead to more confusion, and we thought that the information offered as an image (Figure 1) could be more descriptive, now changed into color. If the reviewer find it important we can change the table to specified each one of the results of figure 1.

4, page12 Line 363-364: What exactly is "many aspects"?

Answer: This information has been detailed at the beginning of the discussion, page 10 line 284-290

5, page12 Line 366-367: What exactly do you mean by 'nAMD patients' needs?

Answer: The sentence has been rewritten to specify more clearly what are patients´ needs: page 12line 392-393

Most of the patients that denote worsening in these questions refer having problems washing and dressing themselves (self-care) o having some problems in their usual activ-ities, as can be seen in table 2. Physical activity decrease significantly during confinement and sedentary time increase. This could accelerate the usual decline in the muscle mass, strength, and function in the older adults affecting specially to activities as "self-care" and "usual activities

Reviewer 2 Report

The manuscript is related to the impact of COVID-19 confinement on the quality of life of patients with age-related macular degeneration. This was a two-wave panel study. The authors claim, and I fully agree, that some patients interrupted the therapy by choice out of fear of SARS-CoV-2 or because it was impossible to reach the hospital. In these ways, the pandemic has added concern to previously distressed patients.

 Several issues need to be clarified:

11.       Line 95

“Patients known to have other ocular comorbidities or other eye conditions that could cause visual

impairment were excluded”. Please provide the details on what basis the patients were excluded. Did any ocular comorbidities occur during the observation period? It is not uncommon for patients treated with intravitreal injections to develop cataract or glaucoma or retinal complications.

22.       Line 205

In the initial survey, the two most frequent health problems were “pain/discomfort” (in 69.23% of patients) and “mobility problems” (38.46%). In the final survey, the two most frequent health problems were “pain/discomfort” and “problems with usual activities” in 66.67% and 48.87% of patients, respectively.

nAMD does not cause pain or discomfort. If those were the most frequent complaints reported by patients, we can be sure they were unrelated to AMD. AMD is a painless vision-impairing retinal disease.

33.       Figure 1 is very difficult to read and be interpreted the given results.

44.       The main conclusion from Finger RP. et al systematic review was: “The introduction of anti-VEGF agents has been associated with a positive impact on patient-relevant outcomes, including a significant reduction in the incidence of blindness and visual impairment by nAMD. Anti-VEGF agents replaced less-effective treatments, improving patient outcomes and broadening the patient population eligible for treatment”. Moreover, the results from multiple randomized clinical trials (RCTs) showed that visual acuity improved over time.  Your results showed the opposite in terms of the gradual deterioration of visual function over time. Please discuss that matter more extensively in the discussion. Is there a rationale for high-cost antiVEGF therapy in the reflection of your results? Please refer this matter to the treatment interruption during a pandemic.

Author Response

REVIEWER 2

The manuscript is related to the impact of COVID-19 confinement on the quality of life of patients with age-related macular degeneration. This was a two-wave panel study. The authors claim, and I fully agree, that some patients interrupted the therapy by choice out of fear of SARS-CoV-2 or because it was impossible to reach the hospital. In these ways, the pandemic has added concern to previously distressed patients.

 Several issues need to be clarified:

  1. Line 95

“Patients known to have other ocular comorbidities or other eye conditions that could cause visual impairment were excluded”. Please provide the details on what basis the patients were excluded. Did any ocular comorbidities occur during the observation period? It is not uncommon for patients treated with intravitreal injections to develop cataract or glaucoma or retinal complications.

Answer: Patients with significant lens opacities, glaucoma, diabetic retinopathy and other comorbidities were excluded in the initial survey. This information has been added in page 3 line 97-99.

Information about cataract surgery between the two surveys has been added in “Study Outcomes” (page 3 line 106).

In the lapse of time between both surveys, 9 eyes of 9 patients developed significant lens opacities and were operated on. No patients developed uncontrolled ocular hyper-tension or other retinal complications.

This information has been added in page 4 line 183-186

  1. Line 205

In the initial survey, the two most frequent health problems were “pain/discomfort” (in 69.23% of patients) and “mobility problems” (38.46%). In the final survey, the two most frequent health problems were “pain/discomfort” and “problems with usual activities” in 66.67% and 48.87% of patients, respectively.

nAMD does not cause pain or discomfort. If those were the most frequent complaints reported by patients, we can be sure they were unrelated to AMD. AMD is a painless vision-impairing retinal disease.

Answer: I agree with the comment.

 Obviously, the high frequency of problems related to pain-disconfort found in our sample is not related to AMD but the questionnaire reflect other health problems of our AMD pa-tients. Pain-disconfort is a frequent complaint in the general population, especially in women (Fig 1)

 We have added this comment in the discussion, page 11 Line 311-314

  1. 33.Figure 1 is very difficult to read and be interpreted the given results.

 Answer: We have change the colour of the figure to improve the comprehension. It could be striking that our AMD patients have better results in EQ5D than obtained in the Spanish 2012 National health Survey. The 2012 Survey was the last national survey that used the EQ5D, that’s why we compared our results with hat Survey. Nevertheless general QoL has improved substancially in later years and it can be observed in other reports.

The next sentence has been rewritten and added the correspondent reference

Although HRQL has improved substantially in last decade in some sense, HRQoL has suffered a setback of 10 years due to the COVID-19 pandemic. [36]

  1. Instituto Nacional de Estadística. https://www.ine.es/prensa/experimental_ind_multi_calidad_vida.pdf. Accessed 8 march 2023.
  2. The main conclusion from Finger RP. et al systematic review was: “The introduction of anti-VEGF agents has been associated with a positive impact on patient-relevant outcomes, including a significant reduction in the incidence of blindness and visual impairment by nAMD. Anti-VEGF agents replaced less-effective treatments, improving patient outcomes and broadening the patient population eligible for treatment”.Moreover, the results from multiple randomized clinical trials (RCTs) showed that visual acuity improved over time.  Your results showed the opposite in terms of the gradual deterioration of visual function over time. Please discuss that matter more extensively in the discussion. Is there a rationale for high-cost antiVEGF therapy in the reflection of your results? Please refer this matter to the treatment interruption during a pandemic.

Answer.- There are several possible reason for this worsening in visual function over time in present work. The patients of our sample were treated for nearly 7 years at the final survey and it has been reported in several real-world observational studies that patients followed for more than 7 years have a vision decrease that can reach a loss of 4.3 letters per year. [25]. Visual loss in the long term has been related to progression to atrophy or suboptimal therapy as was precisely the case of many patients that delay appointments and injections in our sample

(Added in the discussion page 11, line 293-301)

Evans RN, Reeves BC, Phillips D, Muldrew KA, Rogers C, Harding SP, Chakravarthy U; IVAN Study Group. Long-term Visual Outcomes after Release from Protocol in Patients who Participated in the Inhibition of VEGF in Age-related Choroidal Neovascularisation (IVAN) Trial. Ophthalmology. 2020 Sep;127(9):1191-1200. doi: 10.1016/j.ophtha.2020.03.020. Epub 2020 Mar 27. PMID: 32359843; PMCID: PMC7471837.

Round 2

Reviewer 2 Report

All required corrections were incorporated into manuscript. I do not have further comments.